# Plato: Plan to Efficiently Decode for Large Language Model Inference

**Shuowei Jin**[1]*, **Xueshen Liu**[1]*, **Yongji Wu**[2]†, **Haizhong Zheng**[3]‡ **Qingzhao Zhang**[1],
**Atul Prakash**[1], **Matthew Lentz**[4], **Danyang Zhuo**[4], **Feng Qian**[5], **Z. Morley Mao**[1]

[1]University of Michigan    [2]University of California, Berkeley
[3]Carnegie Mellon University    [4]Duke University    [5]University of Southern California

{jinsw,liuxs,qzzhang,aprakash,zmao}@umich.edu
yongji.wu@berkeley.edu
haizhonz@andrew.cmu.edu
{mlentz,danyang}@cs.duke.edu
fengqian@usc.edu

## Abstract

Large language models (LLMs) have achieved remarkable success in natural language tasks, but their inference incurs substantial computational and memory overhead. To improve efficiency, parallel decoding methods like Skeleton-of-Thought (SoT) decompose prompts into sub-problems for concurrent processing. However, these methods significantly compromise answer quality by treating semantically linked sub-problems as independent. We propose Plato, a novel approach that co-designs algorithms and systems for semantic-aware parallel decoding. Plato leverages LLMs to organize sub-problems into a dependency graph based on logical and causal relationships, enabling concurrent decoding of non-dependent nodes while preserving answer coherence and quality. To further enhance efficiency, Plato pipelines planning and node decoding stages, implements a global context cache, and carefully structures node inference prompts to maximize key-value cache reuse and minimize overhead. Our evaluations show that Plato improves throughput by up to 68% over autoregressive decoding while achieving a 40% net win rate in answer quality. Compared to SoT, Plato demonstrates a remarkable 90% quality net-win rate. Ablation studies reveal that our pipeline design improves speedup by 29%, while our KV cache reuse optimization reduces overhead by 75%.

## 1 Introduction

Large language models (LLMs) (Guo et al., 2025; Grattafiori et al., 2024; Jaech et al., 2024) have demonstrated remarkable performance across a wide range of tasks (Nam et al., 2024; Cascella et al., 2023), becoming increasingly important in daily life. A key contributor to the success of LLMs is the transformer architecture, which efficiently models relationships among a large number of context tokens and generates responses token-by-token in an autoregressive manner. However, this autoregressive decoding process inherently limits generation throughput and increases latency, ultimately affecting the user experience (Wang et al., 2024).

Various approaches (Leviathan et al., 2023; Chen et al., 2023; Cai et al., 2024; Fu et al., 2024) have been proposed to improve generation throughput through parallel sampling which generates multiple consecutive tokens at the same time. However, research exploring an orthogonal approach—exploiting parallelization opportunities directly within the **semantic**

---

*Equal contribution.
†Corresponding author: Yongji Wu.
‡Work done while at the University of Michigan.

**space**—remains limited. When humans compose responses to complex questions, they typically begin by creating an outline that defines the answer structure, then develop the full response following this framework (Hadi et al., 2024). This approach enhances logical flow and ensures comprehensive coverage of the topic. Such outlines naturally decompose autoregressive generation tasks into distinct sub-problems in the semantic space, creating opportunities to increase efficiency by generating **independent** components in parallel.

A state-of-the-art contribution in this direction is Skeleton of Thought (SoT) (Ning et al., 2023), which uses prompting techniques to decompose the original problem into independent sub-problems (i.e., the skeleton) and then inferences all sub-problems in parallel. While SoT improves generation efficiency, it often sacrifices response quality because its independence assumption frequently fails in practice. Dependencies commonly exist between sub-problems, particularly when addressing questions requiring strong logical reasoning in domains such as mathematics and programming. For instance, as illustrated in Figure 1, SoT produces five sub-problems where only the first two are truly independent, while the others have dependencies on previous node results to correctly reason through the answer. When these interdependent nodes are inferenced independently, as in SoT's approach, the coherency and correctness of the final answer are compromised.

**Contribution.** This paper presents Plato (Plan to Efficiently Decode), an inference framework that leverages **effective parallelism** in the semantic space to enhance inference efficiency. Unlike previous methods, Plato first prompts the LLM to create a structured plan that decomposes the original problem into nodes (sub-problems) while explicitly identifying the dependencies between these nodes. Subsequently, Plato constructs a Directed Acyclic Graph (DAG) based on these dependencies and exploits the parallelism within the graph structure to perform batched inference. Nodes without unfinished dependencies are submitted to inference asynchronously, enabling parallel execution while maintaining logical coherence. This design ensures high-quality answers while significantly boosting efficiency.

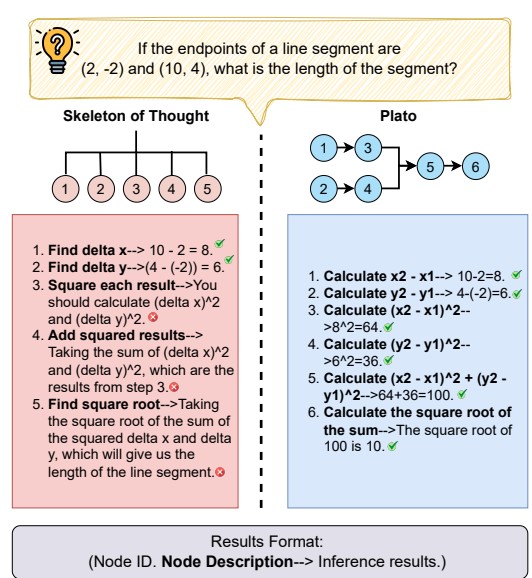

Figure 1: An example to demonstrate the difference between SoT and Plato.

Additionally, we co-designed the inference system to optimize for the Plato workflow. Our system pipelines the planning and node inference stages, significantly reducing latency by ensuring planning doesn't block node inference. Nodes are immediately added to a waiting queue as they are parsed from the planning inference stream. For the nodes in the waiting queue, once all their dependencies are satisfied, they are submitted for inference with results from previously completed nodes as context. To further enhance efficiency, we implement a global context cache and strategically design node inference prompts to maximize key-value cache reuse across requests, substantially reducing computational overhead and improving throughput.

We conducted extensive experiments on nearly 300 questions across 28 skill types to evaluate Plato performance compared with SoT and Autoregressive (AR) inference baselines. Our results show that Plato achieves the optimal balance between speedup and quality. Compared to AR baselines, Plato improves throughput by up to 68% while achieving a 40% net win rate in answer quality. When compared to SoT, Plato demonstrates a remarkable 90% quality net-win rate. Through detailed ablation studies, we show that our pipeline design improves speedup by 29%, while our KV cache reuse optimization reduces overhead by 75%, resulting in just 1.33% total overhead.

In summary, we propose Plato, a novel inference framework that efficiently exploits parallelism within the semantic space to enhance LLM inference efficiency without compromising response quality. Our approach is complementary to existing multi-token consecutive parallel decoding methods (Leviathan et al., 2023; Chen et al., 2023; Cai et al., 2024; Fu et al., 2024), offering an orthogonal optimization dimension. Through the co-design of algorithmic and system components, Plato effectively maximizes throughput while minimizing computational overhead, advancing the state-of-the-art in LLM inference systems.

## 2 Related Work

### 2.1 Efficient LLM Inference Algorithm

**Consecutive multi-token parallel decoding.** Various sampling algorithms have been proposed to enhance generation throughput by parallel sampling multiple consecutive tokens simultaneously, breaking the token-by-token autoregressive generation nature of LLMs. Speculative Decoding algorithms (Leviathan et al., 2023; Chen et al., 2023; Li et al., 2024) employ a smaller model to generate a draft of multiple tokens, which are then verified in parallel by a larger target model, effectively improving the throughput of the sampling process. Similarly, multi-token parallel decoding methods (Cai et al., 2024) decode multiple consecutive tokens simultaneously by fine-tuning the base LLM to generate multiple prediction heads. Lookahead decoding methods (Fu et al., 2024) concurrently extract and verify n-gram tokens in parallel. These approaches collectively improve LLM decoding throughput by generating multiple consecutive tokens at once.

**Parallelism opportunities in semantic space.** Research exploring parallel opportunities directly within the semantic space to enhance inference efficiency remains limited. A notable work in this direction is Skeleton of Thought (SoT) (Ning et al., 2023), which improves LLM generation efficiency by decomposing the original problem into several independent sub-problem nodes that can be decoded in parallel as a batch. However, SoT does not account for potential dependencies among these sub-problem nodes, leading to quality degradation in complex reasoning tasks. Our work further pushes the boundary of this direction by considering necessary dependencies across semantic space, thereby efficiently utilizing semantic parallelism during decoding.

### 2.2 Prompting for Higher Quality

To better exploit the reasoning ability of LLMs, many methods have been proposed to construct more informative and helpful prompts for LLMs to solve complex problems. Chain-of-Thought (CoT) (Wei et al., 2022; Wang et al., 2023) introduces intermediate steps between inputs and outputs as demonstrations in prompts, improving generation quality on complex tasks. Tree-of-Thought (ToT) (Yao et al., 2023) advances LLMs' reasoning capabilities by representing the reasoning process as a tree, with nodes representing different solutions. Graph-of-Thought (GoT) (Besta et al., 2023) extends ToT structure by considering a graph-based approach, introducing aggregation operations on different sub-solutions to further improve reasoning. These methods focus on improving response quality for complex prompts, but ignore generation efficiency; in contrast, our work focus on improving the generation efficiency while not sacrificing quality by only parallelizing independent subtasks by only parallelizing independent subtasks.

## 3 Background

### 3.1 Attention and KV cache

The attention mechanism (Vaswani et al., 2017) is a fundamental component of LLMs that helps models understand relationships between tokens in a sequence. KV cache is an optimization technique that makes LLM inference faster.

The attention mechanism transforms input sequences into query (Q), key (K), and value (V) representations through learned weights: $Q = XW_Q$, $K = XW_K$, $V = XW_V$. It then

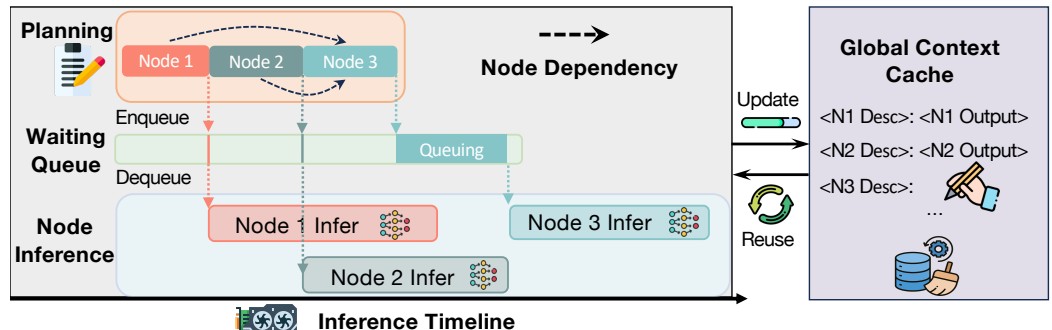

Figure 2: Plato begins with a planning phase where the LLM decomposes the original question into nodes (sub-problems) with their logical dependencies. As each node is generated, it enters a waiting queue. Nodes become eligible for inference when all their dependencies have been satisfied. For example, in the figure, Node 1 and Node 2 have no dependencies, so they are immediately launched for inference upon generation. Node 3, however, depends on both Node 1 and Node 2, so it must remain in the waiting queue until both dependency nodes complete their inference. This dependency-aware scheduling ensures generation quality while maximizing parallel execution opportunities.

computes attention scores as $\text{Attention}(Q, K, V) = \text{softmax}\left(\frac{QK^T}{\sqrt{d_k}}\right)V$, where $d_k$ is a scaling factor for stability.

Without a KV cache, models inefficiently recalculate keys and values for all previous tokens at each generation step. The KV cache optimization stores these computed values, requiring only calculation of the query for the new token, computing its key and value, and updating the cache. This significantly reduces redundant computations during inference.

Additionally, current state-of-the-art LLM inference engines and services (Zheng et al., 2023b; Kwon et al., 2023; OpenAI, 2024b; Anthropic, 2024; Deepseek, 2024) have implemented prompt caching mechanisms. These systems store previously computed KV caches from prior conversations in GPU, CPU memory, or disk (Jin et al., 2024). When subsequent requests contain identical prompt prefixes, the system can reuse these cached KV values, eliminating redundant computation and reducing latency. In Plato design, we leverage this KV cache reuse opportunity to improve overall throughput.

## 3.2 Prefill and Decode Stages in LLM Inference

Current state-of-the-art inference systems (Kwon et al., 2023; Zheng et al., 2023b) divide LLM inference into two distinct stages: the prefill stage and the decode stage. The prefill stage processes the entire input prompt at once, computing and storing the KV cache for all prompt tokens in parallel. This highly parallelizable computation efficiently utilizes GPU resources, though it becomes computation-bound with very long sequences. In contrast, the decode stage generates tokens autoregressively, incrementally updating the KV cache with each new token. This process is primarily memory-bound, as it requires repeatedly loading the KV cache and model weights into GPU memory for each token generation, preventing full utilization of the GPU's computational capacity.

To improve efficiency, systems can batch multiple independent requests together, allowing shared model weights to reduce per-request memory I/O load and enhance inference throughput. The design of Plato exploits these characteristics by identifying and decoding multiple parallelizable sub-problems in a batch, significantly improving throughput compared to traditional autoregressive approaches while maintaining logical coherence between dependent components.

## 4 Design of Plato

Plato co-designs its algorithm and inference system to maximize efficiency. As illustrated in Fig. 2, Plato employs a planning step that exploits parallel opportunities in the semantic space by decomposing the original question into subquestions together with their dependencies on the previous nodes. It then builds a dependency graph on these subquestion nodes and performs parallel batched inference based on this graph structure. To further enhance performance, we design a specialized inference system that improves throughput by overlapping the planning and node inference phases. Additionally, to reduce prefill overhead during node inference, we design a global context cache together with a cache-aware prompting technique that maximizes the utilization of previously generated KV cache.

### 4.1 Core Stages of Plato

Plato operates through two primary stages: a planning stage that decomposes the original question into sub-question nodes and constructs a dependency graph, followed by a node inference stage that exploits the graph's parallel structure to process multiple independent nodes concurrently.

In the planning stage, Plato decomposes complex answering tasks into multiple nodes (subtasks) while preserving their logical dependencies. Prior work (Ning et al., 2023), which treats all subtasks as independent, often fails to capture the intricate relationships between reasoning steps. This can lead to inconsistent or incorrect answers,, as illustrated in Fig. 1. To address this challenge, Plato models subtask relationships as a Directed Acyclic Graph (DAG). During the planning stage, the LLM generates both the content and corresponding dependencies for each subtask node. This dependency-aware approach ensures that information flows correctly through the reasoning process.

In the node inference phase, Plato improves throughput while ensuring quality. Plato enforces logical dependencies by processing nodes in the proper order. A node is launched for inference only after all its dependencies have been satisfied. For instance, if node B depends on node A, Plato first completes node A, then incorporates A's results into B's context before performing inference on B. This structured approach maintains consistency, significantly improving answer quality compared to when all nodes are treated as independent. Additionally, within this graph structure, nodes whose dependencies are satisfied will be launched for inference asynchronously in parallel to improve the system's overall throughput.

### 4.2 Inference System Optimization

Beyond leveraging the inherent parallelism in the graph structure, Plato implements specialized system-level optimizations that exploit the unique characteristics of its workflow to significantly enhance efficiency.

#### 4.2.1 Pipelining Planning and Node Inference

One key insight in Plato is that we can improve throughput and reduce end-to-end latency by pipelining the planning and node inference stages rather than executing them sequentially. Traditional approaches like SoT execute these stages in strict sequence, where node inference only begins after the planning stage is completely finished. This sequential execution creates a substantial bottleneck, as the planning stage introduces non-negligible latency and negatively impacts the system's overall throughput.

To eliminate this bottleneck, we implement a pipelining architecture that overlaps planning with node inference. This is possible due to the autoregressive nature of node generation: earlier nodes cannot depend on later nodes (which haven't been generated yet), while later nodes may depend on earlier ones. This **unidirectional dependency pattern** allows us to begin processing early nodes without waiting for the complete plan.

Our implementation uses real-time token streaming during the planning stage. As tokens are generated, Plato continuously analyzes the output buffer, identifying completed nodes that match our structured format:

> Node ID. Node Content. [Dependency List]

As soon as a complete node is detected in the buffer, it is immediately extracted and dispatched to the waiting queue—without waiting for the entire planning phase to complete. Nodes in the waiting queue whose dependencies have already been satisfied are immediately dequeued and launched for inference, creating a pipeline where planning and node inference overlap to hide latency.

This pipelined execution design transforms a sequential process into a concurrent one, where early nodes in the dependency graph can begin inference while later nodes are still being generated. This results in a substantial reduction in overall latency and more efficient utilization of computational resources throughout the inference process.

### 4.2.2 KV Cache Reuse Optimization

During the node inference stage, each node requires a comprehensive prompt containing the original question and all relevant context information. However, the prefill operations needed to generate KV cache for these lengthy prompts introduce non-negligible computational overhead. To address this challenge, we carefully design the prompt structure combined with a global context cache module. This approach maximizes KV cache reuse across different nodes while minimizing redundant prefill computations by leveraging the prefix caching capabilities of modern LLM inference engines (as described in § 3.1).

We carefully structure the prompt to facilitate efficient KV cache reuse across the node inference requests. It includes four critical components, arranged to maximize caching benefits: (1) the original question, (2) instructions, (3) the global context cache containing all previously generated context, and (4) the specific subtask the current node addresses. By placing the shared contents (1,2,3) at the prefix of the prompt, we ensure the KV cache corresponding to these elements are cached and reused across nodes, significantly reducing redundant computations. The details of the prompt are described in Fig. 6.

Additionally, we design the global context cache module to maintain a consistent context prefix across nodes, maximizing the KV cache reuse opportunities. As each node completes its inference, its output is appended to this cache. An example of the cache content is illustrated in Fig. 2. Subsequent nodes incorporate relevant context from this cache, ensuring accurate information flow and efficient KV cache reuse.

**Counterintuitively, using all previous node results (global context cache) in the context is more efficient than only including dependent nodes' results.** To illustrate, consider a sequence of nodes $A \rightarrow B \rightarrow C \rightarrow D$, with an additional dependency $B \rightarrow D$. The original question and instructions prompts together are represented as $P$. When Plato starts inference on $A$, the prefix prompt will be $P$. After prefill, $P$'s corresponding KV cache will be stored in memory. Once node $A$ finishes, its output $A^*$ is stored in the global context cache, and the system starts Node $B$'s inference. The prompt for Node $B$ has prefix $P+A^*$. Since $P$'s KV cache is already computed, the inference engine loads it and only needs to perform prefill on $A^*$, generating and saving $A^*$'s KV cache for future reuse. Similarly, when node $B$ completes, its output $B^*$ is added to the cache, and node $C$'s prompt becomes $P+A^*+B^*$. The system can reuse the KV cache for $P+A^*$ and only needs to perform prefill computation on $B^*$. When node $D$'s inference begins, its prompt prefix will be $P+A^*+B^*+C^*$, and the system already has KV cache for $P+A^*+B^*$, requiring prefill computation only for $C^*$.

In contrast, if we were to include only the direct dependencies in each node's context (e.g., for node $D$, only including outputs from nodes $B$ and $C$), the prompt would be $P+B^*+C^*$. This approach would significantly reduce KV cache reuse, as KV cache values depend on tokens' positions in the sequence. Since $B^*$ and $C^*$ would appear at different positions compared to their locations in the global context, the system would need to perform additional prefill operations, generating entirely new KV caches. This selective context approach would be less efficient than our global context strategy.

From this analysis, we can see that the global context cache maintains a **consistent prefix order** across nodes, maximizing KV cache reuse opportunities. Each node's output only needs to be prefilled once, after which its KV cache can be reused. In contrast, without our

cache alignment, generating the $i$-th node would typically require re-prefilling all outputs from nodes 1 through $i - 1$, incurring a cumulative overhead of $1 + 2 + \cdots + (N - 1) = O(N^2)$ for $N$ nodes. By ensuring each node's output is added to the cache exactly once and reused thereafter, our approach reduces prefill computation to $O(N)$, significantly minimizing overhead.

# 5 Experiments

We comprehensively evaluate our proposed method aiming to answer the following research questions: (1) How much does Plato affect generation efficiency and quality compared to baseline methods in general? (2) How does Plato perform across different question categories? Which categories benefit most from Plato, and which are less suitable? (3) What benefits does our pipelining planning and node inference optimizations provide? (4) How much overhead does Plato incur?

## 5.1 Experiments Setups

**Dataset.** We evaluate our method using two comprehensive datasets. The primary dataset is Vicuna (Chiang et al., 2023), which contains 80 diverse questions across nine categories. Additionally, we evaluate Plato on WizardLM (Xu et al., 2023), a larger dataset with 218 questions spanning 28 different skill types including academic writing, code debugging, historical analysis, and more. Since the evaluation results on both datasets yield similar conclusions, we focus our discussion on the Vicuna results in the main evaluation section, while WizardLM results are presented in Appendix C.

**Models.** We evaluate three state-of-the-art open-source models of different sizes: Phi-4 (Abdin et al., 2024), Qwen2.5-32B (Team, 2024), and Llama3.1-70B (Meta, 2024). Details regarding the model sizes, API endpoints, and model repositories can be found in Appendix A.

**System Setups.** We build Plato inference system on top of SGLang (Zheng et al., 2023b) v0.4.3 and running inference on two 80 GB NVIDIA A100 GPUs with tensor parallelism.

**Baselines.** We compare our solutions to three different baselines: autoregressive generation (**AR**), chain-of-thought prompting (**CoT**) (Wei et al., 2022), as well as the state-of-the-art semantic-space parallel decoding technique: Skeleton-of-Thought (**SoT**) (Ning et al., 2023). Other consecutive multi-token parallel decoding methods (Leviathan et al., 2023; Chen et al., 2023; Cai et al., 2024; Fu et al., 2024) are orthogonal and complementary to our work as they focus on token-level parallelism rather than semantic-space parallelism, and can be applied to improve the efficiency of either our planning stage or node inference stage.

**Evaluation Metrics.** We evaluate answer quality using the LLM-as-a-judge framework (Zheng et al., 2023a). For each evaluation, the judge is presented with the question and a pair of answers generated by two different decoding techniques on the same model. To mitigate position bias, we swap the order of the two answers and call the judge twice for each comparison. We use GPT-4o-mini (OpenAI, 2024a) as the judge, which achieves similar judging performance as GPT-4o (Hurst et al., 2024) but is 20x cheaper (Tan et al., 2024). We report net win rates (Ning et al., 2023) for each model, calculated as (#wins - #loses) / #questions, where 0% indicates similar answer quality and higher values indicate better answers. For efficiency evaluation, we measure throughput speed-up by calculating the total number of generated tokens divided by the total processing time across all questions. This metric provides a fairer assessment of inference system performance (Kwon et al., 2023; Zhong et al., 2024; Sheng et al., 2023) compared to end-to-end latency, which can be affected by randomly varying answer lengths.

## 5.2 Overall Evaluation

We evaluate different decoding approaches using Autoregressive (AR) generation as our baseline. Figure 3 presents the throughput speed-up and answer quality for each model compared to AR. Our results demonstrate that Plato achieves an optimal balance between quality and efficiency. It not only produces higher quality answers than AR with an average net win rate of 40%, but also delivers inference speed-ups of up to 1.68x.

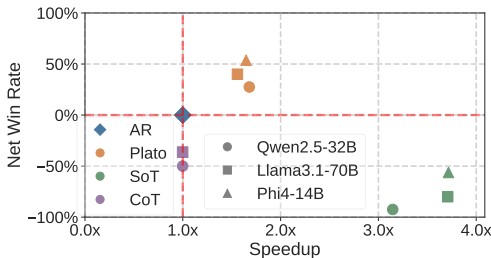

Figure 3: **[Overall Evaluation]:** Answer quality and speed-up compared to normal autoregressive generation (AR) on Vicuna.

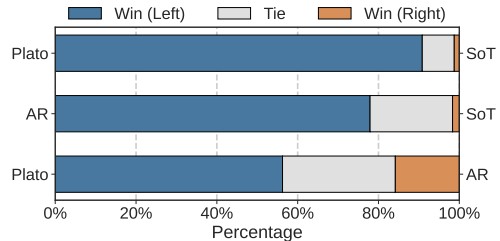

Figure 4: **[Overall Evaluation]:** Quality of answers across all models between different methods on Vicuna.

| Model | Phi4-14B | | Qwen2.5-32B | | Llama3.1-70B | |
|---|---|---|---|---|---|---|
| Method | SoT | Plato | SoT | Plato | SoT | Plato |
| coding | -100% | **14%** | -100% | **-43%** | -86% | **-57%** |
| common-sense | -10% | **80%** | -90% | **50%** | -80% | **30%** |
| counterfactual | -60% | **100%** | -100% | **50%** | -80% | **100%** |
| fermi | -60% | **40%** | -90% | **40%** | -80% | **30%** |
| generic | -40% | **100%** | -90% | **40%** | -90% | **80%** |
| knowledge | -60% | **90%** | -80% | **70%** | -80% | **80%** |
| math | -100% | **-67%** | -100% | -100% | -67% | -67% |
| roleplay | -30% | **20%** | -100% | **50%** | -60% | **50%** |
| writing | -90% | **10%** | -90% | **-20%** | -90% | **10%** |

Table 1: **[Category Breakdown]:** Net win rate of answer quality over AR across all models on each question category.

| Model | Phi4-14B | | Qwen2.5-32B | | Llama3.1-70B | |
|---|---|---|---|---|---|---|
| Method | SoT | Plato | SoT | Plato | SoT | Plato |
| coding | 2.87 | 1.79 | 2.49 | 1.34 | 3.02 | 1.25 |
| common-sense | 3.89 | 1.70 | 3.17 | 2.03 | 3.64 | 2.00 |
| counterfactual | 3.82 | 1.90 | 3.14 | 1.83 | 3.80 | 1.46 |
| fermi | 3.43 | 1.54 | 3.19 | 1.32 | 3.71 | 1.41 |
| generic | 3.87 | 1.54 | 3.39 | 2.01 | 3.87 | 1.75 |
| knowledge | 4.14 | 1.34 | 3.23 | 1.88 | 3.91 | 1.74 |
| math | 2.79 | 1.41 | 2.49 | 1.23 | 2.71 | 1.14 |
| roleplay | 3.95 | 1.50 | 3.15 | 1.57 | 3.76 | 1.54 |
| writing | 3.79 | 1.98 | 3.42 | 1.50 | 4.04 | 1.34 |

Table 2: **[Category Breakdown]:** Speedup over AR across all models on each question category.

In contrast, while SoT achieves the greatest speed-up by processing all sub-question nodes in parallel, its answer quality is significantly worse than AR methods. This quality degradation is expected due to SoT's assumption of complete independence between nodes, which substantially reduces answer coherence and effectiveness.

We also observe that CoT prompting actually degrades performance compared to standard AR generation. This aligns with previous findings that CoT is an emergent ability highly dependent on model scale, primarily benefiting models with approximately 100B parameters or larger (Wei et al., 2022).

Figure 4 further illustrates the win/tie/loss rates across all models when comparing methods pairwise. The results are striking: Plato thoroughly outperforms SoT with a 90% net win rate, while AR achieves a 76% net win rate over SoT. Overall, Plato consistently delivers the highest quality responses, while SoT produces the lowest quality outputs despite its speed advantages.

## 5.3 Category Breakdown

We further analyze performance across different question categories to understand which types of questions are most suitable for Plato and SoT. Table 1 breaks down answer quality and Table 2 shows the speed-up. Both methods are compared against the baseline AR decoding.

From Table 1, we observe that Plato improves quality across most question categories, while SoT decreases quality across all categories. Plato performs worst on tasks with strong sequential dependencies that require holistic outputs, such as coding and math. In these categories, Plato also achieves the least speedup as these tasks offer fewer parallel inference opportunities, with solutions that must be developed step by step.

Upon closer examination of these challenging categories, we find that for coding tasks, Plato tends to produce code fragments with detailed explanations rather than generating complete, cohesive code examples. For math questions, Plato often produces verbose explanations

| Model | Phi4-14B | | | Qwen2.5-32B | | | Llama3.1-70B | | |
|---|---|---|---|---|---|---|---|---|---|
| Metric | TP | PD | IE | TP | PD | IE | TP | PD | IE |
| SoT | 307.81 | 10.86 | 28.34 | 125.59 | 7.86 | 15.98 | 75.10 | 8.21 | 9.15 |
| Plato | 136.46 | 3.87 | **35.26** | 67.05 | 2.01 | **33.36** | 31.57 | 2.37 | **13.32** |

Table 3: **[Compare with SoT]**: Average throughput (TP: token/sec), parallel degree (PD) and inference efficiency (IE: TP/PD) of Plato and SoT over all questions in Vicuna.

| Optimization | Pipeline [seconds] | | | KV Prefill [#tokens] | | |
|---|---|---|---|---|---|---|
| Model | P4 | Q2.5 | L3.1 | P4 | Q2.5 | L3.1 |
| Before Opt. | 16.21 | 17.30 | 49.73 | 4375 | 3495 | 4580 |
| After Opt. | 11.49 | 12.68 | 35.51 | 1165 | 850 | 1048 |
| Reduction | 29% | 27% | 29% | 73% | 76% | 77% |

Table 4: **[Ablation Study]**: Overhead reduction after applying optimization in §4.2.1 and §4.2.2. P4, Q2.5, L3.1 are abbreviations of Phi-4, Qwen2.5 and Llama3.1.

| Model | Phi4-14B | Qwen2.5-32B | Llama3.1-70B |
|---|---|---|---|
| AR | 95.5% | 96.0% | 93.0% |
| SoT | 63.5% | 65.0% | 55.0% |
| Plato | 94.5% | 94.0% | 91.5% |

Table 5: Correctness rate of AR, SoT, and Plato on GSM-8K.

| Model | Phi4-14B | Qwen2.5-32B | Llama3.1-70B |
|---|---|---|---|
| SoT | 2.10× | 2.11× | 2.41× |
| Plato | 1.16× | 1.23× | 1.12× |

Table 6: Speedup of AR and SoT on GSM-8K over AR.

where several steps could ideally be condensed into one. The LLM judge penalizes this redundancy, resulting in lower scores even when the final answer is correct.

To validate this, we conducted additional experiments on the first 200 samples of the GSM-8K dataset (Cobbe et al., 2021). We directly compared the correctness of the numerical results with the ground-truth answers using Deepseek-V3-0324 (DeepSeek-AI, 2024), rather than assessing overall answer quality. As shown in Table 5, while SoT's accuracy drops sharply on math problems, Plato maintains a correctness rate comparable to AR. Furthermore, Table 6 shows that, despite the highly sequential nature of math solutions, Plato can still exploit limited parallelism, achieving up to a 1.23× speedup over AR.

Additionally, Plato excels in both quality and efficiency for questions requiring breadth rather than sequential reasoning. For these tasks, Plato can generate more comprehensive answers in parallel, incorporating more examples and covering broader aspects of the topic, thus simultaneously improving both quality and speedup.

We provide three case studies in Appendix E for a deeper analysis of the differences in generation results between AR, SoT, and Plato.

## 5.4 System-level Inference Efficiency Comparison

To understand the overall system-level inference efficiency, we compare Plato with SoT in Table 3, reporting throughput, parallel degree, and inference efficiency.

For Plato, we define the parallelism degree as the total number of nodes divided by the maximum path length in the DAG graph generated during the planning phase. For SoT, the parallelism degree equals the number of nodes in the skeleton. We also introduce an inference efficiency (IE) metric, calculated as speed-up divided by parallelism degree, to evaluate how efficient the inference is independent of parallelism benefits. A higher IE indicates a more optimized system.

The results show that Plato significantly outperforms SoT in terms of inference efficiency, with improvements ranging from 24% to 108%. This demonstrates the effectiveness of Plato's approach in co-designing algorithms and systems to achieve superior efficiency.

## 5.5 Ablation Study on System Optimization

We evaluate the impact of our system optimizations on both inference speedup and KV cache prefill overhead.

**Pipeline Optimization.** The left side of Table 4 shows the inference latency before and after applying our pipeline design that overlaps planning and node inference stages. This optimization yields a consistent 27-29% speedup (latency reduction) across all models, demonstrating the effectiveness of our pipeline approach.

**KV Cache Reuse.** A significant overhead in Plato comes from prefill computations during multi-node inference. The right side of Table 4 shows that our KV cache reuse strategy reduces the number of additional prefill tokens by approximately 75% across all models. This substantial reduction in prefill overhead contributes significantly to the overall efficiency of Plato while maintaining answer quality.

Additionally, we report the exact overhead percentage caused by prefilling additional tokens. Thanks to the KV cache reuse optimization in §4.2.2, Plato introduces minimal latency overhead—less than 1.6% of the total generation time across all tested models. The detailed overhead analysis for different models is presented in Table 10.

## 6 Conclusion

In this paper, we propose Plato, which pushes the boundary of semantic space parallelism to enhance inference efficiency. Plato codesigns the inference algorithm together with the system, significantly improving both answer quality and generation speed. Our comprehensive evaluations demonstrate that Plato achieves an optimal balance between speedup and quality, with a 90% quality net-win rate over SoT while improving throughput by up to 68% compared to autoregressive baselines. Our pipeline design improves speedup by 29%, and KV cache reuse optimization reduces overhead by 75%, demonstrating the effectiveness of our algorithm-system co-design approach.

Looking ahead, Plato's principles can extend to multi-agent systems, complex reasoning, and collaborative AI workflows. By modeling dependencies between sub-problems and leveraging semantic space parallelism, Plato advances LLM inference efficiency and effectiveness, opening new research avenues at the intersection of algorithmic design and systems optimization.

## Acknowledgments

We would like to thank our anonymous reviewers for their valuable comments and feedback. This work was supported in part by NSF under grants CMMI-2038215, CNS-2321532, CNS-2323174, OAC-2503010, CNS-2402696, CNS-2238665 and National AI Institute for Edge Computing Leveraging Next Generation Wireless Networks, Grant–2112562.

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

| Model | Hugging Face Path or API Model Name |
|---|---|
| GPT-4o-mini (OpenAI, 2024a) | gpt-4o-mini |
| Phi4-14B (Abdin et al., 2024) | microsoft/phi-4 |
| Qwen2.5-32B (Team, 2024) | Qwen/Qwen2.5-32B-Instruct |
| Llama3.1-70B (Meta, 2024) | meta-llama/Meta-Llama-3.1-70B-Instruct |

Table 7: Hugging Face or API Model Name of all used models.

## A    Models Details

We summarizes Hugging Face Hubs' repos of the open-source models and the GPT model we used in Table 7. We run the open-source models using SGlang (Zheng et al., 2023b) v0.4.3 with 2xA100 80G.

For all LLM-as-a-judge (FastChat) (Zheng et al., 2023a) evaluations, we use GPT-4o-mini as the judge LLM with the API model name *gpt-4o-mini*.

## B    Prompt Details

**Plan Generation Prompt**    We demonstrate our plan generation prompt for Plato in Figure 5.

To ensure the generation of node dependency graphs that are both informative and straightforward to parse, we define the expected output format at the beginning of the prompt. This specification serves to align the Large Language Model (LLM) output with our parsing requirements, thereby ensuring that the generated skeletons are easy to parse. Then we provide guidelines to instruct the LLM to generate a comprehensive and cohesive plan while maximizing the parallelism of decoding. To complement this specification, we provide an illustrative example that serves as one-shot learning instance.

We also observe that models like Qwen2.5-32B and Phi-4 follow the instructions well. Meanwhile, Llama3.1-70B produces unstable outputs from time to time.

**Node Inference Prompt**    We demonstrate our node inference prompt for Plato in Figure 6.

The node inference prompt helps the LLM focus on the subtask it is assigned to, and provides detailed and in-depth answer to it, making the overall response better.

The template begins by reminding the LLM of the main question, which helps to keep the answer focused and on track. Then we state the requirements for answering the subtask. Following this, we append the description and results of all finished subtasks at this point. Because the generation waits until its dependent subtasks are finished, the relevant information has been included in the context. Instructions and contexts are only prefilled once, as their KV cache will be reused repeatedly during parallel node inference. After the context, we add the description of the subtask.

**Planning Prompt of Plato.**

```
<|im_start|>system
```
Decompose the given problem into subtasks with dependencies, optimizing for parallel execution (shortest critical path). Format each subtask as (each line is a subtask):

> `<id>`. Subtask description.`[dependency_ids]`

Guidelines:

- Conciseness: Keep descriptions short and action-oriented (e.g., "Calculate X", "Summarize Y").
- Dependencies: Only specify dependencies if a subtask requires outputs from others.
- Readability: Ensure the final concatenated answer flows logically (e.g., analysis follows data generation).
- Parallelism: Maximize independent subtasks; minimize critical path length.
- Comprehensiveness: Explore any relevant aspects of the given problem in parallel.
- Structure: Ensure a clear logical flow between subtasks and avoid redundancy and fragmentation.
- Completion: End with a synthesis subtask to reach a *very short* and cohensive final answer.

Example:
Problem: "Compare the average GDP growth of Country A and B over 5 years, then recommend investments.
Response:

> 1. Fetch GDP data for Country A (2019-2023).`[]`
> 2. Fetch GDP data for Country B (2019-2023).`[]`
> 3. Calculate average GDP growth for Country A.`[1]`
> 4. Calculate average GDP growth for Country B.`[2]`
> 5. Compare growth rates and identify higher-performing country.`[3,4]`
> 6. Recommend investment strategy based on comparison.`[5]`

Note: Subtasks 1-2 and 3-4 can run in parallel; 1/2-3/4-5-6 form the critical path. `<|im_end|>`
```
<|im_start|>user
```
**[Insert Question Here]** `<|im_end|>`
```
<|im_start|>assistant
```

Figure 5: Plan generation prompt of Plato.

**Node Inference Prompt of Plato.**

```
<|im_start|>system
```
You are an AI assistant that can answer questions.`<|im_end|>`
```
<|im_start|>user
```
Your team is solving the question: **[Insert Question Here]**
Your task is to complete one specific subtask in a processing plan.
You are an expert solving a subtask as part of a larger workflow. Follow these rules:

- Conciseness: Answer in 1-3 sentences if possible.
- Depth: Include key insights/calculations (e.g., formulas, reasoning, examples).
- Consistency: Consider the logic flow from previous results and the question and be cohensive.

Previous results: **[Global Contexts]**
Directly state your answer to the subtask: **[Subtask Description]**.`<|im_end|>`
```
<|im_start|>assistant
```

Figure 6: Node inference prompt of Plato.

| Model | Phi4-14B | | | | | | | |
|---|---|---|---|---|---|---|---|---|
| Method | SoT | | | | Plato | | | |
| Result | Win | Tie | Lose | Net win rate | Win | Tie | Lose | Net win rate |
| Academic Writing | 0 | 2 | 2 | -50% | 3 | 0 | 1 | 50% |
| Art | 0 | 3 | 2 | -40% | 4 | 1 | 0 | 80% |
| Biology | 2 | 2 | 2 | 0% | 4 | 1 | 1 | 50% |
| Chemistry | 0 | 0 | 2 | -100% | 2 | 0 | 0 | 100% |
| Code Debug | 0 | 3 | 7 | -70% | 6 | 4 | 0 | 60% |
| Code Generation | 0 | 2 | 16 | -89% | 4 | 9 | 5 | -6% |
| Common-Sense | 1 | 6 | 2 | -11% | 8 | 1 | 0 | 89% |
| Complex Format | 1 | 0 | 11 | -83% | 3 | 2 | 7 | -33% |
| Computer Science | 0 | 3 | 12 | -80% | 10 | 3 | 2 | 53% |
| Counterfactual | 1 | 6 | 1 | 0% | 8 | 0 | 0 | 100% |
| Economy | 0 | 1 | 4 | -80% | 3 | 2 | 0 | 60% |
| Entertainment | 2 | 2 | 1 | 20% | 4 | 1 | 0 | 80% |
| Ethics | 1 | 2 | 3 | -33% | 4 | 2 | 0 | 67% |
| History | 0 | 1 | 3 | -75% | 2 | 2 | 0 | 50% |
| Law | 2 | 1 | 2 | 0% | 5 | 0 | 0 | 100% |
| Literature | 0 | 1 | 4 | -80% | 3 | 2 | 0 | 60% |
| Math | 1 | 4 | 15 | -70% | 8 | 10 | 2 | 30% |
| Medicine | 0 | 1 | 4 | -80% | 3 | 2 | 0 | 60% |
| Multilingual | 0 | 1 | 6 | -86% | 3 | 2 | 2 | 14% |
| Music | 1 | 1 | 3 | -40% | 4 | 1 | 0 | 80% |
| Physics | 0 | 3 | 2 | -40% | 1 | 3 | 1 | 0% |
| Reasoning | 1 | 4 | 8 | -54% | 8 | 4 | 1 | 54% |
| Roleplay | 0 | 3 | 3 | -50% | 4 | 2 | 0 | 67% |
| Sport | 0 | 1 | 4 | -80% | 3 | 2 | 0 | 60% |
| Technology | 0 | 2 | 4 | -67% | 4 | 2 | 0 | 67% |
| Toxicity | 1 | 2 | 1 | 0% | 4 | 0 | 0 | 100% |
| TruthfulQA | 1 | 3 | 1 | 0% | 3 | 2 | 0 | 60% |
| Writting | 0 | 2 | 16 | -89% | 13 | 2 | 3 | 56% |
| **Overall** | 15 | 62 | 141 | **-58%** | 131 | 62 | 25 | **49%** |

Table 8: Answer quality of SoT and Plato compared with auto-regressive (AR) across different categories on WizardLM. The answers are generated by Phi-4.

## C Evaluation on WizardLM

In addition to the Vicuna dataset, we also evaluate Plato on a larger dataset, WizardLM (Xu et al., 2023). It contains 218 questions covering a wide range of skills, such as academic writing, code-debug, history, etc. We use standard autoregressive (AR), Skeleton-of-Thought (SoT), and Plato to answer these questions and evaluate the output using LLM-as-a-judge (FastChat) (Zheng et al., 2023a). We present the results in Table 8 and Table 9 with detailed breakdown across different categories.

Across all the subjects and models, Plato achieves positive total net win rate over AR while SoT suffers from significant quality drop, showing that Plato effectively maintains answer quality while improving the token generation speed.

| Model | Qwen2.5-32B | | | | | | | |
|---|---|---|---|---|---|---|---|---|
| Method | SoT | | | | Plato | | | |
| Result | Win | Tie | Lose | Net win rate | Win | Tie | Lose | Net win rate |
| Academic Writing | 0 | 0 | 4 | -100% | 0 | 4 | 0 | 0% |
| Art | 1 | 1 | 3 | -40% | 2 | 3 | 0 | 40% |
| Biology | 0 | 1 | 5 | -83% | 4 | 1 | 1 | 50% |
| Chemistry | 0 | 0 | 2 | -100% | 1 | 1 | 0 | 50% |
| Code Debug | 0 | 0 | 10 | -100% | 4 | 6 | 0 | 40% |
| Code Generation | 0 | 0 | 18 | -100% | 2 | 4 | 12 | -56% |
| Common-Sense | 1 | 3 | 5 | -44% | 4 | 4 | 1 | 33% |
| Complex Format | 0 | 1 | 11 | -92% | 0 | 5 | 7 | -58% |
| Computer Science | 0 | 0 | 15 | -100% | 5 | 6 | 4 | 7% |
| Counterfactual | 0 | 1 | 7 | -88% | 5 | 2 | 1 | 50% |
| Economy | 0 | 0 | 5 | -100% | 4 | 1 | 0 | 80% |
| Entertainment | 0 | 1 | 4 | -80% | 2 | 1 | 2 | 0% |
| Ethics | 1 | 2 | 3 | -33% | 5 | 1 | 0 | 83% |
| History | 0 | 0 | 4 | -100% | 2 | 2 | 0 | 50% |
| Law | 0 | 0 | 5 | -100% | 1 | 4 | 0 | 20% |
| Literature | 0 | 1 | 4 | -80% | 1 | 3 | 1 | 0% |
| Math | 0 | 3 | 17 | -85% | 2 | 12 | 6 | -20% |
| Medicine | 0 | 0 | 5 | -100% | 3 | 2 | 0 | 60% |
| Multilingual | 0 | 1 | 6 | -86% | 2 | 4 | 1 | 14% |
| Music | 0 | 2 | 3 | -60% | 3 | 2 | 0 | 60% |
| Physics | 0 | 1 | 4 | -80% | 1 | 3 | 1 | 0% |
| Reasoning | 0 | 4 | 9 | -69% | 6 | 5 | 2 | 31% |
| Roleplay | 0 | 0 | 6 | -100% | 5 | 1 | 0 | 83% |
| Sport | 0 | 0 | 5 | -100% | 2 | 1 | 2 | 0% |
| Technology | 0 | 3 | 3 | -50% | 5 | 1 | 0 | 83% |
| Toxicity | 0 | 1 | 3 | -75% | 2 | 2 | 0 | 50% |
| TruthfulQA | 0 | 0 | 5 | -100% | 2 | 2 | 1 | 20% |
| Writting | 2 | 1 | 15 | -72% | 11 | 5 | 2 | 50% |
| **Overall** | 5 | 27 | 186 | **-83%** | 86 | 88 | 44 | **19%** |

Table 9: Answer quality of SoT and Plato compared with auto-regressive (AR) across different categories on WizardLM. The answers are generated by Qwen2.5-32B.

| Model | Phi4-14B | Qwen2.5-32B | Llama3.1-70B |
|---|---|---|---|
| Avg. prefill time [s] | 0.13 | 0.19 | 0.47 |
| Avg. total time [s] | 11.49 | 12.68 | 35.51 |
| Overhead | 1.17% | 1.52% | 1.32% |

Table 10: Overhead of Plato after applying KV cache reuse optimization. Time is averaged over all questions in Vicuna.

## D  Overhead Analysis

We present the latency overhead results across different models in Table 10.

## E  Case Studies

We present three case studies to compare answers from AR, SoT, and Plato generated by different models in details.

### E.1  Comparison of Answers Generated by Phi4-14B

**Question:**  What are the main differences between Python and JavaScript programming languages?

**(Concise) AR Answer:**  Python and JavaScript are both popular programming languages, but they have distinct characteristics and use cases. Here are some of the main differences between the two:

- *Design and Syntax*: - **Python**: Known for its clean and readable syntax, Python uses indentation to define code blocks, which enhances readability. It follows a philosophy of simplicity and minimalism. - **JavaScript**: Uses curly braces '{}' to define code blocks and semicolons ';' to end statements. Its syntax can be more verbose and less intuitive for beginners compared to Python.
- *Typing*: - **Python**: Dynamically typed, but with optional type hints (introduced in Python 3.5), allowing for some static type checking. - **JavaScript**: Also dynamically typed, but with the addition of TypeScript, a statically typed superset that provides optional static type checking.
- *Execution Environment*: - **Python**: Primarily a server-side language, but also used for scripting, data analysis, and machine learning. It is executed in a variety of environments including standalone scripts, web servers (e.g., Django, Flask), and data science platforms. - **JavaScript**: Originally designed for web browsers to add interactivity to web pages. It is now widely used on both the client-side (browsers) and server-side (Node.js) with the rise of full-stack JavaScript development.
- *Concurrency*: - **Python**: Uses a Global Interpreter Lock (GIL) that can be a bottleneck for CPU-bound multithreading. However, it supports asynchronous programming with 'asyncio' and can leverage multiprocessing for parallelism. - **JavaScript**: Utilizes an event-driven, non-blocking I/O model, making it highly suitable for asynchronous operations. It uses Promises and 'async/await' to handle asynchronous code more easily.
- *Use Cases*: - **Python**: Commonly used in web development, data science, artificial intelligence, scientific computing, automation, and scripting. - **JavaScript**: Primarily used for front-end web development, but also for back-end development with Node.js. It is essential for building interactive web applications.
- *Libraries and Frameworks*: - **Python**: Rich ecosystem with libraries like NumPy, Pandas, TensorFlow, and frameworks like Django and Flask. - **JavaScript**: Extensive library ecosystem with tools like React, Angular, Vue.js for front-end development, and Express.js, Koa.js for server-side development.

- *Community and Support*: - Both languages have large, active communities and extensive documentation, but the focus of their communities differs based on their primary use cases.

These differences highlight the distinct roles and strengths of Python and JavaScript, making them suitable for different types of projects and development needs.

**(Concise) SoT Answer:**

- *Language type. key concept*: Python is primarily an interpreted, high-level programming language focused on code readability and simplicity, often used for scripting and rapid application development. JavaScript is a lightweight, interpreted or JIT-compiled programming language with first-class functions, primarily used for client-side scripting in web browsers.
- *Execution model. key concept*: Python is an interpreted language, typically executed line-by-line, which can make debugging easier but may result in slower execution. JavaScript, originally designed for web browsers, is also interpreted but often compiled just-in-time (JIT) by modern engines like V8, enhancing performance.
- *Syntax differences. key concept*: Python emphasizes readability with its use of indentation to define code blocks, while JavaScript uses curly braces and semicolons, allowing for more flexible syntax styles.
- *Use cases. key concept*: Python is widely used in data science, machine learning, and scientific computing due to its extensive libraries like NumPy and Pandas, while JavaScript is predominantly used for web development, enabling dynamic and interactive web pages through frameworks like React and Angular.
- *Typing system. key concept*: Python is dynamically typed with optional type hints, allowing more flexibility, while JavaScript is also dynamically typed but has introduced TypeScript, a statically typed superset, to enhance type safety and development experience.
- *Concurrency model. key concept*: Python uses the Global Interpreter Lock (GIL) which limits execution to one thread at a time, whereas JavaScript employs an event-driven, non-blocking I/O model with an event loop, allowing for efficient handling of asynchronous operations.
- *Libraries and frameworks*: Python boasts a rich ecosystem with libraries like NumPy and frameworks like Django for web development, while JavaScript is renowned for its extensive libraries such as React and frameworks like Angular and Vue.js, catering primarily to front-end development.
- *Community and support*: Both Python and JavaScript have large, active communities, but Python is often praised for its emphasis on readability and simplicity, which attracts beginners, while JavaScript's community is heavily focused on web development and rapidly evolving with frameworks like React and Node.js.
- *Performance characteristics. key concept*: Python is generally slower than JavaScript due to its interpreted nature and dynamic typing, while JavaScript, especially with the V8 engine, is optimized for high performance in web environments.
- *Browser support. key concept*: JavaScript is natively supported by all web browsers, making it essential for client-side web development, while Python is not directly supported in browsers but can be used server-side or through transpilers like Brython or Pyodide for client-side scripting.

**(Concise) Plato Answer (dependent nodes are in []):**

- *Identify core use cases of Python* []: Python is primarily used for web development, data analysis, artificial intelligence, scientific computing, and automation scripting. Its simplicity and readability make it popular in data science and machine learning. Python's extensive libraries, like NumPy, Pandas, and TensorFlow, support its use in scientific and analytical applications. Additionally, frameworks such as Django and Flask facilitate web development, while its versatility allows use in automation and backend server applications.

- *Identify core use cases of JavaScript* []: JavaScript is primarily used for web development, enabling dynamic and interactive content on websites through client-side scripting. It is essential for front-end development, interacting with web browsers via the Document Object Model (DOM), and is also used in back-end development with platforms like Node.js. Additionally, JavaScript is utilized in building mobile applications through frameworks like React Native and in desktop applications using Electron.

- *Analyze syntax differences between Python and JavaScript* []: Python and JavaScript have distinct syntax differences. Python uses indentation to define code blocks, promoting readability, whereas JavaScript relies on curly braces '{}'. Python's syntax is generally more concise and clean, with dynamic typing and simpler function definitions, while JavaScript supports both functional and object-oriented programming with a more complex syntax including semicolons and explicit type declarations. Additionally, Python uses 'def' to define functions, whereas JavaScript uses the 'function' keyword or arrow functions ('=¿').

- *Compare performance characteristics of Python and JavaScript* []: Python and JavaScript have distinct performance characteristics due to their design and use cases. Python is typically slower than JavaScript in execution speed because it is an interpreted language, often requiring optimization or using extensions like Cython for performance-critical applications. In contrast, JavaScript, especially with modern engines like V8, is designed for fast, efficient execution in web browsers and server environments (Node.js), benefiting from just-in-time (JIT) compilation. While JavaScript can handle asynchronous operations efficiently, particularly with its non-blocking I/O model, Python requires additional frameworks (such as asyncio) to achieve similar asynchronous capabilities.

- *Evaluate ecosystem and library support for Python* []: Python boasts a robust ecosystem with extensive library support, particularly strong in data science, machine learning, and scientific computing. Libraries such as NumPy, Pandas, and TensorFlow provide powerful tools for numerical analysis, data manipulation, and deep learning. For web development, frameworks like Django and Flask offer comprehensive solutions for building complex web applications, and libraries such as Requests facilitate network operations. Python's package manager, pip, simplifies the installation and management of these libraries, ensuring a rich and versatile development environment. Overall, Python's ecosystem is well-integrated and continuously evolving, making it a preferred choice for a wide range of applications.

- *Evaluate ecosystem and library support for JavaScript* []: JavaScript has a robust ecosystem and extensive library support, primarily driven by its use in web development. Key libraries and frameworks include React, Angular, and Vue.js for front-end development, which enable the creation of dynamic user interfaces. For back-end development, Node.js allows JavaScript to run on the server-side, supported by libraries like Express.js for building web applications. Additionally, Electron provides tools for building cross-platform desktop applications, and React Native extends JavaScript's reach to mobile app development. The npm (Node Package Manager) registry further bolsters JavaScript's ecosystem, offering one of the largest collections of open-source libraries and tools, facilitating extensive community support and rapid development.

- *Compare learning curves and community support for both languages* []: Python and JavaScript both have extensive community support and resources, but their learning curves differ due to syntax and use cases. Python is often considered easier for beginners due to its simple, readable syntax and emphasis on code readability, making it ideal for learning programming fundamentals. It has a large community, with comprehensive documentation and numerous educational resources available, which aids in learning. On the other hand, JavaScript's learning curve can be steeper due to its more complex syntax, including asynchronous operations and prototypal inheritance, which may require a deeper understanding of web technologies. . . .

- *Synthesize findings into a concise comparison of Python and JavaScript* [3,4,5,6,7]: Python and JavaScript are both versatile languages with distinct core use cases. Python excels in data analysis, scientific computing, and artificial intelligence, supported by libraries like NumPy and TensorFlow, and is favored for its readability and simplicity. JavaScript, on the other hand, is primarily used for web development, enabling dynamic content on

websites and supporting both front-end and back-end development through frameworks like React, Node.js, and React Native. . . .

**Findings:** We find that the AR answer provides a clear, structured overview of Python and JavaScript differences but lacks synthesis and in-depth technical insights (e.g., JIT compilation, browser support). SoT answer is more comprehensive but is overly granular and fragmented. The Plato answer systematically breaks down key aspects (syntax, performance, ecosystems) and synthesizes them into a cohesive comparison, avoiding redundancy while maintaining depth.

### E.2 Comparison of Answers Generated by Qwen2.5-32B

**Question:** Can you explain the basics of quantum computing?

**(Concise) AR Answer:** Certainly! Quantum computing is a type of computing that uses quantum mechanics to process information. It leverages the principles of superposition and entanglement to perform operations on data.

- *Qubits*: Unlike classical bits, which are the basic unit of information in classical computing and can be either 0 or 1, quantum bits or qubits can exist in multiple states simultaneously thanks to superposition. This means a qubit can be 0, 1, or any quantum superposition of these states.
- *Superposition*: This principle allows a quantum system to be in multiple states at the same time. For example, a qubit can be in a state that is a combination of both 0 and 1 until it is measured. The act of measurement collapses the qubit into one of its possible states, which could be 0 or 1.
- *Entanglement*: This is a phenomenon where quantum particles become interconnected and the state of one (no matter the distance) can depend on the state of another. For instance, if two qubits are entangled, the state of one qubit instantly influences the state of the other. This property is crucial for many quantum computing operations and can be used to perform certain computations much faster than classical computers.
- *Quantum Gates*: In classical computing, logic gates (like AND, OR, NOT) manipulate bits to perform logical functions. Quantum gates manipulate qubits in similar ways but use quantum operations that can take advantage of superposition and entanglement. These gates can perform complex operations on multiple qubits simultaneously, which is one of the reasons quantum computers have the potential to be significantly more powerful than classical computers for certain tasks.
- *Quantum Algorithms*: Algorithms like Shor's algorithm for factoring large numbers and Grover's algorithm for searching unsorted databases are examples of algorithms that exploit quantum mechanical phenomena to solve problems much faster than any classical algorithm could.
- *Challenges*: Quantum computing faces significant challenges, such as decoherence (loss of quantum behavior due to interaction with the environment) and error rates, which are higher than those in classical computing. Quantum error correction techniques are being developed to address these issues.

Quantum computers are particularly promising for solving problems in cryptography, optimization, and simulation of quantum systems, which are currently infeasible for classical computers. However, practical large-scale quantum computers are still in the early stages of development and research.

**(Concise) SoT Answer:**

- *Quantum bits (qubits).*: Qubits are the fundamental units of quantum information, analogous to classical bits, but capable of existing in multiple states simultaneously due to the principle of superposition.
- *Superposition principle. key concept*: The superposition principle allows qubits to exist in multiple states simultaneously, represented as a combination of both 0 and 1, enabling

quantum computers to process vast amounts of information more efficiently than classical computers.

- *Entanglement phenomenon. key concept*: Entanglement occurs when qubits become correlated in such a way that the state of one (whether it's 0 or 1) can depend on the state of another, no matter the distance between them, allowing for instantaneous state changes that are crucial for quantum computing operations.

- *Quantum interference. key concept*: Quantum interference allows constructive or destructive interference of probability amplitudes, enabling quantum algorithms to amplify correct answers and cancel out wrong ones, thus optimizing computational outcomes.

- *Quantum gates. key concept*: Quantum gates are operations performed on qubits to manipulate their states, analogous to classical logic gates, but they can also create superposition and entanglement, enabling quantum computations.

- *Quantum algorithms. key concept*: Quantum algorithms, such as Shor's and Grover's, leverage quantum properties to solve problems more efficiently than classical algorithms, offering exponential speedups for specific tasks.

- *Decoherence challenges. key concept*: Decoherence refers to the loss of quantum behavior in qubits due to interaction with the environment, leading to errors in quantum computations and posing a significant challenge in maintaining quantum state integrity.

- *Quantum supremacy. key concept*: Quantum supremacy refers to the point where a quantum computer can solve a problem that a classical computer cannot solve within a reasonable time frame, demonstrating the superior computational power of quantum systems.

**(Concise) Plato Answer (dependent nodes are in []):**

- *Define quantum bits (qubits)* **[]**: Quantum bits, or qubits, are the fundamental units of quantum information, analogous to classical bits but capable of existing in a superposition of states, representing both 0 and 1 simultaneously until measured. This property stems from the principles of quantum mechanics, enabling qubits to perform multiple computations at once, potentially offering significant speedups for certain tasks.

- *Explain superposition* **[1]**: Superposition in quantum computing allows a qubit to exist in multiple states simultaneously, not just binary 0 or 1 as in classical computing. Mathematically, a qubit can be represented as a linear combination $\alpha|0\rangle + \beta|1\rangle$, where $\alpha$ and $\beta$ are complex numbers such that $|\alpha|^2 + |\beta|^2 = 1$, encapsulating the probabilities of measuring the qubit in state $|0\rangle$ or $|1\rangle$.

- *Describe entanglement* **[1]**: Entanglement is a quantum phenomenon where qubits become interconnected such that the state of one (whether it's 0 or 1) cannot be described independently of the state of another, no matter the distance separating them. This interconnection allows operations on one qubit to instantaneously affect another, enabling complex quantum algorithms and secure quantum communication protocols.

- *Introduce quantum gates* **[1]**: Quantum gates are the basic building blocks of quantum circuits, analogous to classical logic gates but operating on qubits. These gates manipulate the state of qubits using unitary transformations, enabling the creation of superpositions, entanglements, and interference, which are crucial for quantum algorithms. Examples include the Hadamard gate, which creates superposition, and the CNOT gate, which generates entanglement.

- *Summarize quantum algorithms* **[2,3,4]**: Quantum algorithms leverage the principles of superposition, entanglement, and quantum gates to perform specific computational tasks more efficiently than classical algorithms. Notable examples include Shor's algorithm for factoring large integers exponentially faster than the best-known classical algorithms, and Grover's algorithm for unstructured search problems, providing a quadratic speedup. These algorithms exploit quantum parallelism and interference to achieve their speedups.

- *Compare quantum computing with classical computing* **[5]**: Quantum computing fundamentally differs from classical computing in its ability to leverage superposition and entanglement, allowing qubits to exist in multiple states simultaneously and become interconnected in ways that classical bits cannot. This enables quantum computers to perform certain operations in parallel and process complex computations more efficiently

than classical computers for specific tasks, such as factoring large numbers and searching unstructured databases.

- *Outline potential applications of quantum computing* [5]: Quantum computing has the potential to revolutionize several fields by solving complex problems more efficiently. Key applications include cryptography, where quantum computers could break current encryption methods but also enable quantum key distribution for secure communication; optimization in logistics and finance, where quantum algorithms can find optimal solutions faster; and simulation of quantum systems in chemistry and materials science, enabling the development of new drugs and materials.

**Findings:** We find that all three methods give a comprehensive introduction to quantum computing, while Plato uses concrete examples to elaborate concepts. Plato also provides a clear and structured logical flow with comparison and summarization, making the whole answer coherent, while the answer from SoT is fragmented.

### E.3 Comparison of Answers Generated by Llama3.1-70B

**Question:** How many times does the average human blink in a lifetime? Try to explain your answer. Your explanation should take the reader through your reasoning step-by-step.

**(Concise) AR Answer:**

- *Step 1: Determine the average blink rate per minute*: Research suggests that the average person blinks around 12-15 times per minute. Let's use a conservative estimate of 13 blinks per minute.
- *Step 2: Calculate the number of blinks per hour*: There are 60 minutes in an hour, so we can multiply the average blink rate per minute by 60 to get the number of blinks per hour: 13 blinks/minute $\times$ 60 minutes/hour = 780 blinks/hour.
- *Step 3: Calculate the number of blinks per day*: Assuming an average of 16 hours of wakefulness per day (accounting for 8 hours of sleep), we can multiply the number of blinks per hour by the number of hours spent awake: 780 blinks/hour $\times$ 16 hours/day = 12,480 blinks/day.
- *Step 4: Calculate the number of blinks per year*: There are 365.25 days in a year (accounting for leap years). We can multiply the number of blinks per day by the number of days in a year: 12,480 blinks/day $\times$ 365.25 days/year $\approx$ 4,555,200 blinks/year.
- *Step 5: Estimate the average human lifespan*: According to the World Health Organization (WHO), the average human lifespan is approximately 72 years (based on global data from 2019).
- *Step 6: Calculate the total number of blinks in a lifetime*: Finally, we can multiply the number of blinks per year by the average human lifespan: 4,555,200 blinks/year $\times$ 72 years $\approx$ 327,971,200 blinks/lifetime.
- *Conclusion*: Therefore, the average human blinks approximately **327,971,200 times** in a lifetime, assuming an average blink rate of 13 blinks per minute, 16 hours of wakefulness per day, and an average human lifespan of 72 years. Keep in mind that this is a rough estimate and actual blink rates may vary from person to person. Nonetheless, this calculation should provide a reasonable ballpark figure.

**(Concise) SoT Answer:**

- *Average blinks per minute*: According to various studies, the average person blinks around 12-15 times per minute. This rate can vary depending on factors such as attention, environment, and eye health, but 12-15 blinks per minute is a commonly cited estimate.
- *Minutes per day*: Assuming an average person sleeps for 8 hours a day, that leaves 16 hours of wakefulness, which translates to 960 minutes per day.
- *Days per year*: Assuming a non-leap year, there are 365 days in a year, and for a leap year, there are 366 days, so we can take the average of the two, which is approximately 365.25 days per year.

- *Years in lifetime*: Assuming an average human lifespan of around 80 years, we can use this as a rough estimate for our calculation.
- *Calculate total blinks*: To calculate the total number of blinks, we multiply the average blinks per minute (point 1) by the minutes per day (point 2) and then by the days per year (point 3) and finally by the years in a lifetime (point 4). This gives us a preliminary estimate of total blinks in a lifetime.
- *Consider blinking variations*: We must account for variations in blinking rates due to factors such as age, activity level, and environment, which can significantly impact the average blink rate.
- *Estimate lifetime blinks*: Based on these calculations, the estimated total number of blinks in a lifetime is approximately 415 million, assuming an average lifespan of 80 years. However, this is a rough estimate and actual numbers may vary significantly from person to person.

**(Concise) Plato Answer (dependent nodes are in []):**

- *Determine the average human lifespan* **[]**: Based on global data, the average human lifespan is approximately 72 years (70 years for males and 74 years for females), according to the World Health Organization (2019). This estimate takes into account factors such as mortality rates, life expectancy, and population demographics. I will use this value as a baseline for subsequent calculations.
- *Calculate the average number of blinks per minute* **[]**: The average number of blinks per minute is approximately 15-20 blinks per minute (bpm). This rate can vary depending on factors like age, environment, and activity level, but 17.5 bpm is a commonly cited average. This rate will serve as the basis for further calculations to determine the total number of blinks in a lifetime.
- *Estimate the average number of minutes spent awake per day* **[]**: Based on typical adult sleep patterns, I estimate the average number of minutes spent awake per day to be around 960 minutes (16 hours). This assumes 7-9 hours of sleep per night and 24 hours in a day.
- *Calculate the average number of blinks per day* **[2,3]**: Based on the estimated average number of minutes spent awake per day (960 minutes) and the assumed average blink rate (17.5 blinks per minute), the average number of blinks per day can be calculated as: 960 minutes/day * 17.5 blinks/minute = 16,800 blinks/day.
- *Calculate the average number of blinks per year* **[4]**: To calculate the average number of blinks per year, we can multiply the average number of blinks per day (16,800 blinks/day) by the number of days in a year (365.25, accounting for leap years). This results in: 16,800 blinks/day * 365.25 days/year ≈ 6,133,800 blinks/year.
- *Multiply the average number of blinks per year by the average human lifespan* **[1,5]**: Multiplying the average number of blinks per year (6,133,800 blinks/year) by the average human lifespan (72 years), we get: 6,133,800 blinks/year * 72 years ≈ 441,652,800 blinks in a lifetime. This represents the estimated total number of blinks an average human would make over the course of their life.

**Findings:** We find that all three answers are close. However, Plato answer delivers the most rigorous and structured reasoning process, leveraging logic dependency to ensure coherence and transparency. For SoT answer, although the value makes sense, doesn't have a cohesive explanation process, which directly retrieves the final results.

