# OpenReview forum: "Plato: Plan to Efficient Decode for Large Language Model Inference"
_colmweb.org/COLM/2025/Conference — COLM 2025_

### Official Review · Reviewer_5cB7 · 2025-05-07

**Rating:** 7
**Confidence:** 4
**Ethics Flag:** 1

**Summary:**

This work point out that the current parallel decoding methods divide the problem to subproblem for parallel processing. This work suggests that such parallel deocing may be harmful for the answer because the subproblems are solved independently. Thereofore, this work propose Plato  for semantic-aware parallel decoding.

**Questions To Authors:**

N/A

**Reasons To Accept:**

* This work proposes to build a relation for the subproblems to improve the performance, which mitigation the limitation of previous parallel decoding.
* The work conducted a sufficient experiment to prove the effectiveness of Plato. This work evaluates Plato with different datasets, different models. These experiments support the effectiveness of Plato.
* The paper is easy to read.

**Reasons To Reject:**

* The Plato needs more time because some decoding may have to wait for another.
* There may be more datasets used for evaluation. For example, there is no math dataset evaluation.
* The work maximum model size 70B, which may be larger for further validation.

---

> ### Author Response · Authors · 2025-06-03
>
> We thank the reviewer for the detailed feedback and for recognizing the strengths of our work. Regarding your concerns:
> * There is no actual additional waiting time overhead introduced by Plato’s design. Even in the worst case, where each node sequentially depends on the previous one, there is always a node being decoded in Plato’s pipeline. As a result, Plato’s decoding throughput in such fully sequential scenarios falls to the same level as standard autoregressive decoding, with no additional penalty.
>
> * Thank you for bringing the concerns about our evaluation dataset to our attention. Our initial experiments intentionally followed setups aligned with previous work (such as SoT [1]), focusing on diverse question benchmarks that include nearly 300 questions across 28 types for comprehensive evaluation. To thoroughly assess Plato’s accuracy on math problems, we conducted additional experiments using the GSM-8K dataset. Specifically, we compared the accuracy performance of standard autoregressive decoding (AR), SoT, and Plato using the first 200 samples from GSM-8K, testing Phi-4 14b, Qwen2.5-32b, and Llama3.1-70b models. We employed Deepseek-V3-0324 to directly compare numeric results with ground truth answers provided by GSM-8K. Results are summarized below:
>
> **Correctness rate:**
> | Method          | Phi-4 14b | Qwen2.5 32b | Llama3.1 70b |
> | --------------- | --------- | ----------- | ------------ |
> | Autoregressive        | 95.5%     | 96.0%       | 93.0%        |
> | SoT            | 63.5%     | 65.0%       | 55.0%        |
> | Plato        | 94.5%     | 94.0%       | 91.5%        |
>
> **Speedup over autoregressive:**
> | Method          | Phi-4 14b | Qwen2.5 32b | Llama3.1 70b |
> | --------------- | --------- | ----------- | ------------ |
> | SoT            | 2.10×     | 2.11×       | 2.41×        |
> | Plato         | 1.16×     | 1.23×       | 1.12×        |
>
>
> Because the steps of math problems are highly sequential, parallel opportunity is very limited, and thus the speedup of Plato over autoregressive is moderate. Still, the 1.17 × average speed‑up across the three models shows that Plato can exploit the limited parallel opportunities between steps while achieving the same level of accuracy as the autoregressive baselines, which demonstrates the effectiveness of Plato’s design. We also note that the quality of SoT is quite limited, which aligns with our evaluation in the paper.
>
>
> * We appreciate your comment on the model size. Due to resource limitations, we only evaluated models up to 70B, which is a common evaluation setting within academic resources.
>
> Thank you again for your thoughtful review and valuable suggestions. We hope that our response and revisions have adequately addressed your concerns and questions. We are more than glad to answer any further questions.
>
> [1] Ning, Xuefei, et al. "Skeleton-of-thought: Large language models can do parallel decoding." ICLR’2024

---

> > ### Comment · Reviewer_5cB7 · 2025-06-04
> >
> > Thanks for the precious response. The above experiments have addressed my concerns. Therefore, I will keep the positive score.

---

### Official Review · Reviewer_JH4e · 2025-05-08

**Rating:** 8
**Confidence:** 3
**Ethics Flag:** 1

**Summary:**

The authors introduce “Plato”, a novel approach that integrates algorithmic and system-level innovations for semantic-aware parallel decoding. Plato utilizes LLMs to construct a dependency graph that captures logical and causal relationships among sub-problems. This enables concurrent decoding of independent nodes while maintaining answer coherence and quality. In addition, to further optimize efficiency, their approach pipelines the planning and decoding stages, employs a global context cache and carefully designs node prompts to maximize key-value (KV) cache reuse and reduce computational overhead. Empirical evaluations show promising results against standard autoregressive decoding and against Skeleton-of-Thought (SoT), also a parallel decoding technique that enhances efficiency by breaking prompts into sub-problems for simultaneous processing.

**Reasons To Accept:**

- The authors introduce a novel approach for decomposing prompts into sub-problems that carefully preserves the semantic dependencies between them. Unlike methods such as Skeleton-of-Thought (SoT), which treat each sub-problem as entirely independent and therefore risk fragmenting the overall coherence of the response, their method explicitly models logical and causal relationships among sub-components. This semantic-aware design allows for parallel processing without sacrificing the consistency or quality of the final answer.

- By maintaining these dependencies, the approach not only avoids the degradation in answer coherence seen in existing parallel decoding strategies but also enhances computational efficiency through selective concurrency. Empirical results demonstrate that this method achieves superior answer quality compared to prior models, highlighting its effectiveness in balancing performance and accuracy.

**Reasons To Reject:**

From my perspective, this work provides substantial value to the research community. I believe it merits acceptance.

---

> ### Author Response · Authors · 2025-06-03
>
> We thank the reviewer for the positive feedback and for recognizing the novelty and significance of our work.
>
> We appreciate your detailed summary of our contributions and your recognition of how our approach maintains semantic coherence while leveraging parallelism. Your understanding of our motivation to preserve logical and causal relationships for robust and accurate multi-node decoding aligns perfectly with our core goals.
>
> We also appreciate your recognition of our empirical results, which highlight Plato’s ability to outperform standard autoregressive decoding and SoT in terms of both quality and efficiency.
>
> Thank you again for your thoughtful review and for highlighting the strengths and potential impact of our work. We are open to further discussion if you have any additional questions.

---

### Official Review · Reviewer_qy58 · 2025-05-12

**Rating:** 6
**Confidence:** 3
**Ethics Flag:** 1

**Summary:**

The core idea of structuring parallel decoding through semantic-aware dependency graphs is intuitive and addresses a known limitation in existing approaches like SoT. The combination of planning with logical/causal relationships and efficient decoding is interesting. This paper is well written.

**Questions To Authors:**

1. What is the principle to select these two datasets for the experiment? Is there any reason except that there are different categories?

**Reasons To Accept:**

1. This paper co-designs planning with system optimization to improve efficiency, while demonstrating better answer quality than other baselines.
2. The motivation of the proposed method is well articulated and the paper is well written.

**Reasons To Reject:**

1. Coding and math are two important tasks, but the current framework doesn't perform well on these two types of tasks.
2. The planning should be highly related to the models and the quality of planning should be very important in order to apply Plato. Plato mainly does experiments on two instruction tuning datasets Vicuna and WizardLLM. It's skeptical how Plato can solve more complex and realistic tasks that need more complex reasoning.

---

> ### Author Response · Authors · 2025-06-03
>
> We sincerely thank the reviewer for the thoughtful and detailed feedback, and we are pleased to address your concerns below.
>
> * Thank you for raising the concerns regarding coding and math tasks. As discussed in the paper, Plato achieves superior performance in general tasks such as writing, common-sense reasoning, and knowledge-based question answering, which represent the main use cases for over 51% of users in the survey [1]. The relatively weaker performance of Plato on math tasks in our initial evaluation arises because the structured planning approach leads to lengthier, more detailed explanations, which do not align with the preferences of LLM-based judges. However, this does not imply that Plato generates incorrect math answers.
> To thoroughly assess Plato’s accuracy on math problems, we conducted additional experiments using the GSM-8K dataset. Specifically, we compared the accuracy performance of standard autoregressive decoding (AR), SoT, and Plato using the first 200 samples from GSM-8K, testing Phi-4 14b, Qwen2.5-32b, and Llama3.1-70b models. We employed Deepseek-V3-0324 to directly compare numeric results with ground truth answers provided by GSM-8K. Results are summarized below:
>
> **Correctness rate:**
> | Method          | Phi-4 14b | Qwen2.5 32b | Llama3.1 70b |
> | --------------- | --------- | ----------- | ------------ |
> | Autoregressive        | 95.5%     | 96.0%       | 93.0%        |
> | SoT            | 63.5%     | 65.0%       | 55.0%        |
> | Plato        | 94.5%     | 94.0%       | 91.5%        |
>
> **Speedup over autoregressive:**
> | Method          | Phi-4 14b | Qwen2.5 32b | Llama3.1 70b |
> | --------------- | --------- | ----------- | ------------ |
> | SoT            | 2.10×     | 2.11×       | 2.41×        |
> | Plato         | 1.16×     | 1.23×       | 1.12×        |
>
> Because the steps of math problems are highly sequential, parallel opportunity is very limited, and thus the speedup of Plato over autoregressive is moderate. Still, the 1.17x average speedup across three models shows that Plato is able to exploit any possible parallel opportunities between steps while achieving the same level of accuracy comparing with autoregressive baselines, which demonstrates the effectiveness of Plato’s design. We also notice that the quality of SoT is quite limited, which aligns with our evaluation in the paper.
>
> * Our initial experiments intentionally followed setups aligned with previous work (such as SoT [1]), focusing on diverse question benchmarks, which include nearly 300 questions across 28 types for a comprehensive evaluation. Additionally, as demonstrated through the GSM-8K evaluation, Plato maintains competitive accuracy even in mathematically oriented tasks. Regarding coding tasks, we acknowledge that Plato is less suitable. Plato’s decomposition strategy breaks coding tasks into smaller units interspersed with natural language explanations, making it challenging to aggregate these parts into a cohesive and functional code snippet. Autoregressive methods, which produce a single continuous coding output, remain better suited to such use cases. We will clarify this limitation explicitly in the revised manuscript.
>
> Thank you once again for your insightful feedback. We hope that our response and revisions have adequately addressed your concerns and questions. We are more than glad to answer any further questions.
>
>
> [1] Infodocket. Report: Close encounters of the AI kind: The increasingly human-like way people are engaging with language models. 2025.
>
> [2] Ning, Xuefei, et al. "Skeleton-of-thought: Large language models can do parallel decoding." ICLR’2024

---

> > ### Comment · Reviewer_qy58 · 2025-06-09
> >
> > Thank you for your response. I'll maintain my score.

---

### Official Review · Reviewer_NYGo · 2025-05-12

**Rating:** 6
**Confidence:** 4
**Ethics Flag:** 1

**Summary:**

This paper presents a novel method for prompt decomposition into sub-tasks. The authors evaluate its performance in terms of speed and quality against direct autoregressive decoding (AR) and Skeleton-of-Thought (SoT). The method features parallel processing capabilities and a queuing architecture for asynchronous sub-task processing, while maintaining awareness of inter-task dependencies. The implementation includes an optimization using a global KV cache for the attention layer. Experimental results demonstrate Plato's superior performance across multiple metrics compared on a selection of four LLMs.

**Questions To Authors:**

Figure 3: please move (outside?) and change the legend; it's really hard to understand.

**Reasons To Accept:**

- The global KV cache is an interesting optimization approach.
- The paper is well structured and follows a logical flow.
- The experimental setup is designed well and thoroughly explained.
- The "Related Work" section provides comprehensive context and embeds the paper in the research environment.
- The proposed method demonstrates significant improvements over both the AR and SoT approaches on the mentioned benchmark datasets.

**Reasons To Reject:**

- The name "Plato" (derived from "Plan to Efficiently Decode") appears somewhat arbitrary and could benefit from a more intuitive justification.
- In the "Design of Plato" section:
  - The pipeline architecture description contains redundant explanations of the concurrent approach.
  - The prompting strategy for problem decomposition deserves more detailed discussion.
  - Better referencing to appendix. The prompt templates should be integrated into the main body of the paper with comprehensive explanations, rather than being demoted to the appendix.
- Some smaller literal and  grammatical mistakes.
- The experimental analysis would benefit from addressing limitations such as the small sample size. Maybe, including confidence intervals or a discussion of reproducibility.
- The consideration of the complexity of the prefill computation needs more explanation. O(N^2) in line 271 is not so clear to me. I also was wondering about prefill for nested graphs.

---

> ### Author Response · Authors · 2025-06-03
>
> We sincerely thank the reviewer for the thoughtful and detailed feedback, and we are pleased to address your concerns below.
>
> * **Regarding the naming of Plato:** Thank you for highlighting your concerns about the name "Plato." The name derives partially from the phrase "Plan to Efficiently Decode," emphasizing our key contributions in task decomposition and efficiency. Additionally, it references the philosopher Plato, known for advocating structured reasoning, paralleling our approach's emphasis on logical coherence via dependency graphs. We will explicitly clarify this rationale in the final version.
> * **On the Design Section, Prompt Templates, and Minor Errors:** We appreciate your feedback on the “Design” section and will incorporate these suggestions in the final version. Additionally, we will carefully proofread and correct the minor grammatical and literal errors identified throughout the manuscript.
> * **Experimental Analysis and Reproducibility:** Thank you for suggesting improvements regarding reproducibility. We will add a detailed discussion section covering the specific choice of the temperature parameter used in our experiments and release the implementation code to facilitate easier reproduction. Our initial experiments intentionally followed setups aligned with previous work (such as SoT [1]), focusing on diverse question benchmarks, which include nearly 300 questions across 28 types for a comprehensive evaluation. Additionally, we have included updated GSM8K results to provide broader empirical validation on math datasets, as shown below.
> We compared the performance between standard autoregressive decoding, SoT, and Plato on math problems using the first 200 samples from the test set of GSM-8K. We tested Phi4-14b, Qwen2.5-32b, and Llama3.1-70b and used Deepseek-V3-0324 to check correctness. Because GSM-8K provides ground truth answers to math problems, we asked Deepseek to directly compare the numeric results and determine if the answer in the response was correct. The results are shown in the following table:
>
> **Correctness rate:**
> | Method          | Phi-4 14b | Qwen2.5 32b | Llama3.1 70b |
> | --------------- | --------- | ----------- | ------------ |
> | Autoregressive        | 95.5%     | 96.0%       | 93.0%        |
> | SoT            | 63.5%     | 65.0%       | 55.0%        |
> | Plato        | 94.5%     | 94.0%       | 91.5%        |
>
> **Speedup over autoregressive:**
> | Method          | Phi-4 14b | Qwen2.5 32b | Llama3.1 70b |
> | --------------- | --------- | ----------- | ------------ |
> | SoT            | 2.10×     | 2.11×       | 2.41×        |
> | Plato         | 1.16×     | 1.23×       | 1.12×        |
>
>
> Because the steps of math problems are highly sequential, parallel opportunities are very limited, and thus the speedup of Plato over autoregressive is moderate. Still, the 1.17x average speedup across three models shows that Plato is able to exploit any possible parallel opportunities between steps while achieving the same level of accuracy comparing with autoregressive baselines, which demonstrates the effectiveness of Plato’s design. We also notice that the quality of SoT is quite limited, which aligns with our evaluation in the paper.
>
> * **Clarifying Prefill Overhead (O(N²)):** We appreciate your suggestion to clarify the complexity of the prefill computation. The O(N²) complexity arises because, without our optimized global KV cache, generating the i-th node typically requires redundant prefills of previously generated nodes (1 through i-1). Thus, a naive approach incurs cumulative redundant overhead (1 + 2 + ... + N-1), leading to an O(N²) complexity. In contrast, our design avoids this by using incremental KV caching.
>
> * **Regarding nested graphs:** In our framework, it is inherently impossible for an earlier node to depend on a later node and thus form a nested graph, due to the autoregressive nature of generation. During the planning phase, each node is generated in a sequential order—when an earlier node is generated, information about any subsequent (future) nodes is not yet available and therefore cannot be referenced. To further ensure system robustness, we have implemented explicit checks in our system to prevent any scenario where an earlier node depends on a later node, thus eliminating the possibility of deadlocks. Empirically, we have not observed any instances of such invalid dependencies occurring. We will clarify these points further in the revised manuscript.
>
> * **Improving Figure 3:**  Thank you for pointing out the clarity issues with Figure 3. We will revise this figure to improve readability, moving the legend externally and enhancing annotations, as suggested.
>
> We thank the Reviewer for the insightful feedback. We hope that our response and revisions have adequately addressed your concerns and questions. We are more than glad to answer any further questions.
>
> [1] Ning, Xuefei, et al. "Skeleton-of-thought: Large language models can do parallel decoding." ICLR’2024

---

> > ### Comment · Reviewer_NYGo · 2025-06-10
> >
> > Thanks for your elaborations. I am not familiar with the corpus but are math-tasks really sequential? (e.g., 1*3+2*4...)
> > Furthermore, I wonder if the quality drop in Plato (1-2 % points) is related to the few occasions where it uses parallel execution? Again, I would love to see more detailed evaluations of those 200 data points.

---

> > ### Author Response · Authors · 2025-06-11
> >
> > Thank you for the follow‑up questions. We appreciate the opportunity to clarify our observations on math tasks and the small quality gap you noted. Please see our responses below:
> >
> >
> > **Are math questions fully sequential?**
> > We do not claim that math questions are fully sequential. However, compared to other general question types, math problems often require more rigorous step-by-step reasoning, which reduces the degree of exploitable parallelism. This trend is also reflected in Table 2 of our paper, where the average speedup for math tasks is noticeably lower than for other genres.
> >
> > **Is the speedup due to parallelism at the cost of correctness?**
> > To address this, we performed a deeper analysis by filtering for only the correct answers (a large enough sample for stability) and then computing the average speedup:
> > - **Phi-4 14b:** 1.156×
> > - **Qwen2.5 32b:** 1.223×
> > - **Llama3.1 70b:** 1.125×
> >
> > These results show that the observed speedup is **not** attributable to incorrectly generated answers. The efficiency gain appears consistently across all correct nodes with parallel execution.
> >
> >
> > **What causes the small quality drop?**
> > The minor quality drop is not due to parallel execution itself, but rather to occasional inaccuracies introduced by the more complex plan-then-decode pipeline. For example, consider the following GSM8K math problem:
> > > **Question:** Josh decides to try flipping a house. He buys a house for 80,000 and then puts in 50,000 in repairs. This increased the value of the house by 150%. How much profit did he make?
> > The **Plato** response is:
> > > 1. Calculate the initial cost of the house: The initial cost of the house is the the purchase price which is 80,000.
> > > 2. Calculate the total repair costs: The total repair costs are 50,000.
> > > 3. Compute the total investment (initial cost + repair costs): 130,000. Depends on 1, 2
> > > 4. Determine the increase in value due to repairs (50,000 * 150%): 75,000. Depends on 2
> > > 5. Calculate the new value of the house (initial cost + increase in value): 130,000 + 75,000 = 205,000. Depends on 1,4
> > > 6. Calculate the profit (new value of the house - total investment): \$205,000 - \$130,000 = \$75,000. Depends on 3, 5
> >
> > Here, the error arises **not** from parallel execution, but from a planning-stage inaccuracy: when computing the increased value of the house, the plan incorrectly uses 50,000 (repair cost) instead of 80,000 (purchase price). This kind of occasional error can disrupt the reasoning flow and thus affect the final result in math tasks.
> >
> >
> > **Summary:**
> > - Math tasks show lower parallelism than other genres due to their inherently sequential logic.
> > - The observed speedup is stable and **not** due to errors from parallel execution.
> > - The occasional minor drop in quality comes from the complexity of the plan-decode process, not from the parallelization itself.
> > We hope these clarifications address your concerns, and we thank you again for your insightful questions.

---

> > > ### Comment · Reviewer_NYGo · 2025-06-11
> > >
> > > Thanks for these additional explanations including the analysis of where/how things go wrong, occasionally. I'll stick with my positive evaluation.

---

### Decision · Program_Chairs · 2025-07-08

**Decision:**

Accept

**Comment:**

This paper presents Plato, a training-free method for efficient LLM inference using semantic-aware parallel decoding. It builds a dependency graph to preserve logical structure and applies system optimizations to improve speed and output quality.

Reviewers initially found the idea promising and the experiments solid, though they raised concerns about performance on math and coding tasks, some unclear technical details, and generalization. The authors provided detailed responses and additional experiments, which addressed most concerns.

Overall, I recommend acceptance. The method is practical and well-supported. The final version should clearly state limitations and include more implementation details in the main text.